# Prognostic Value of Myosteatosis and Albumin–Bilirubin Grade for Survival in Hepatocellular Carcinoma Post Chemoembolization

**DOI:** 10.3390/cancers16203503

**Published:** 2024-10-17

**Authors:** Kittipitch Bannangkoon, Keerati Hongsakul, Teeravut Tubtawee, Natee Ina

**Affiliations:** Department of Radiology, Faculty of Medicine, Prince of Songkla University, Hat Yai 90110, Songkhla, Thailand; hkeerati@medicine.psu.ac.th (K.H.); tteeravu@medicine.psu.ac.th (T.T.); inatee@medicine.psu.ac.th (N.I.)

**Keywords:** muscle change, chemoembolization, myosteatosis, sarcopenia, TACE, prognostic score, body composition, hepatocellular carcinoma, ALBI grade, liver cancer

## Abstract

In patients with hepatocellular carcinoma (HCC), predicting survival after treatment is crucial for informed decision-making. This retrospective study investigated how muscle health (myosteatosis) and liver function (the albumin–bilirubin grade) influence survival outcomes in patients undergoing transarterial chemoembolization (TACE), resulting in the development of the Myo-ALBI prognostic score. The Myo-ALBI score was found to be a more accurate tool for predicting survival compared to existing methods. This study highlights the potential of the Myo-ALBI score to help clinicians personalize treatments and improve patient outcomes.

## 1. Introduction

Hepatocellular carcinoma (HCC) is a severe complication of chronic liver disease, particularly liver cirrhosis, that often leads to poor outcomes [1]. It ranks as the third most common cause of cancer-related deaths globally [2]. The Barcelona Clinic Liver Cancer (BCLC) staging system is the most widely recognized framework for guiding treatment in HCC patients [3,4]. For intermediate-stage HCC (BCLC-B), transarterial chemoembolization (TACE) is the standard treatment [4]. While TACE has been shown to improve survival in BCLC-B HCC, it can also result in the deterioration of liver function, particularly in patients with a high tumor burden and limited hepatic reserve, which leads to unfavorable outcomes [5,6].

The Child–Pugh classification, commonly used to assess liver function reserve in cirrhosis, includes subjective elements, like ascites and hepatic encephalopathy [7]. As a more objective alternative, the albumin–bilirubin (ALBI) grade, based on serum albumin and total bilirubin levels, has been proposed for evaluating liver function reserve across HCC stages [7]. The ALBI grade is categorized into three grades: grade 1 (≤−2.6), grade 2 (>−2.6 to ≤−1.39), and grade 3 (>−1.39) [7]. A recent study by Zhao et al. highlighted the superiority of the ALBI grade over Child–Pugh classification in predicting overall survival (OS) in HCC patients undergoing TACE [8]. Given that serum albumin and bilirubin levels are affected by various non-hepatic factors, there is a growing need for more precise liver function evaluation models. Recent developments include scoring systems that integrate the ALBI grade for a more thorough assessment of hepatic function and improved prognosis prediction in HCC [9,10,11].

Skeletal muscle density (SMD) at the third lumbar (L3) vertebra, which is measured in Hounsfield units (HU) via computed tomography (CT) images, serves as a muscle quality indicator. Low SMD, or myosteatosis, signifies pathological fat accumulation in muscle tissue that can potentially lead to inflammation, insulin resistance, and adverse outcomes in cancer treatment due to inflammatory adipokines from intermuscular fat [12,13,14,15,16]. Recent findings have linked myosteatosis with a diminished response to TACE and reduced survival in HCC patients [17].

However, the combined prognostic effect of myosteatosis and the ALBI grade on clinical outcomes in HCC remains largely unknown. Therefore, this study aimed to assess the prognostic efficacy of preoperative myosteatosis and the ALBI grade and develop a robust prognostic score based on these factors to efficiently predict long-term outcomes for patients with unresectable HCC.

## 2. Materials and Methods

### 2.1. Ethics

This study was reviewed and approved by the Ethics Committee and Institutional Review Board of the Faculty of Medicine at Prince of Songkla University and Songklanagarind Hospital (approval number 65-379-7-1). The study protocol adhered to the standards set forth in the Declaration of Helsinki and current ethical guidelines. Due to the retrospective nature of the study and the use of anonymous clinical data, informed consent was waived. We accessed the medical records of patients between November 2022 and December 2023. The data were anonymized at the point of collection to ensure confidentiality and compliance with ethical standards.

### 2.2. Study Population

Data were retrospectively collected from patients with unresectable HCC who underwent their first session of TACE at a tertiary referral center in Thailand between January 2009 and December 2020. The inclusion criteria were as follows: (1) age over 18 years; (2) preserved liver function as defined by a Child–Pugh classification of A–B; (3) HCC diagnosed through an arterial enhancing liver mass on a background of chronic liver disease or cirrhosis, confirmed via imaging, or histopathological findings in accordance with the American Association for the Study of Liver Disease guidelines [18]; and (4) HCC classified as BCLC stage A, B, or C, including subsegmental or segmental portal vein tumor thrombosis. The exclusion criteria were applied to patients who had undergone surgery, ethanol injection, ablation, systemic chemotherapy, or radiotherapy prior to TACE, as well as patients with missing imaging data, a history of spontaneous tumor rupture, or concurrent cancer.

### 2.3. Data Collection

Data collection included baseline patient characteristics, such as age, sex, and body mass index (BMI), skeletal muscle index (SMI), and SMD, as well as etiologies of liver disease, namely, hepatitis B virus (HBV) or hepatitis C virus (HCV) infection and alcohol consumption. Clinical histories encompassing diabetes, hypertension, cardiovascular disease, and chronic kidney disease were compiled. Tumor characteristics, including the number and size of HCC nodules and BCLC staging, were documented. Liver function tests, which were conducted before TACE, measured the levels of total bilirubin, albumin, platelet count, and the International Normalized Ratio (INR) along with serum alpha-fetoprotein (AFP). Assessments included the Child–Pugh classification and the ALBI grade. The ALBI grade was calculated using the following formula: ALBI grade = (log10 bilirubin × 0.66) + (albumin × −0.085), where bilirubin was measured in µmol/L and albumin in g/L [7]. The up-to-seven criteria were defined as the sum of the diameter of the largest tumor (in centimeters) and the number of tumors [11]. The primary outcome of the current study was OS, which was calculated from the initial TACE session date to the date of death from any cause or the censoring date of 30 June 2023. Progression-free survival (PFS) was calculated from the date of the initial TACE session to the date of radiological progression according to mRECIST criteria, death from any cause, or the censoring date of 30 June 2023, whichever occurred first. Survival data for the enrolled patients were sourced from the national statistical data provided by the Thailand National Death Register.

### 2.4. Skeletal Muscle Index (SMI) and Skeletal Muscle Density (SMD)

SMI and SMD were derived from CT images at the L3 level, which were archived as Digital Imaging and Communications in Medicine (DICOM) data. CT scans conducted within one month prior to the first TACE session were selected for SMI and SMD measurements. Body composition analysis of all DICOM data was performed using in-house software developed on MATLAB R2022b (academic use) (The MathWorks, Natick, MA, USA, https://www.mathworks.com/products/matlab.html (accessed on 1 December 2023)) and Python 3.6.13 (Python Software Foundation, https://www.python.org/downloads/release/python-3613/ (accessed on 1 December 2023)). The Anaconda distribution (Anaconda, Inc., Austin, TX, USA) was used for Python environment management. This software formed the basis of the measurement model [19]. The SMI was calculated by dividing the cross-sectional muscle area by the patient’s height squared (cm^2^/m^2^). According to the study by Fujiwara et al. in 1257 HCC patients, low SMI (or sarcopenia) is defined as less than 36.2 cm^2^/m^2^ for males and less than 29.6 cm^2^/m^2^ for females [13]. SMD values of the abdominal and back muscles were determined from pixel areas with attenuation values between −29 and +150 HU. Males with an HU value less than 44.4 HU and females with an HU value less than 39.3 were classified as having low SMD or the presence of myosteatosis [13].

### 2.5. TACE Procedure and Treatment Schedule

Conventional TACE was performed by experienced interventional radiologists with a minimum of 10 years of experience. The procedure began by assessing the right common femoral artery, followed by selective catheterization of the arterial branches feeding the tumor. Chemoembolization was conducted using a mixture of Lipiodol (4–16 mL) combined with doxorubicin hydrochloride (5–50 mg) or mitomycin-C (10–20 mg). This was followed by embolizing the tumor-feeding arteries using gelatin sponges. Four weeks post-procedure, patients underwent a liver CT scan according to the TACE protocol. Additional TACE sessions were scheduled as needed every 6–12 weeks based on residual or recurrent tumor presence and the patient’s hepatic function reserve.

### 2.6. Statistical Analysis

We characterized the study population using descriptive statistics. Continuous variables were expressed as mean ± standard deviation if normally distributed or median with interquartile range (IQR) for skewed distributions. We reported categorical variables as absolute numbers and relative frequencies. Between-group comparisons utilized Student’s *t*-test for continuous data and chi-square or Fisher’s exact test for categorical variables.

Survival analysis employed Kaplan–Meier estimates, with curve comparisons conducted via log-rank tests. We identified prognostic factors for overall survival (OS) using Cox proportional hazards models, first in univariate analysis and then in multivariate models for significant variables. We also examined associations between skeletal muscle density (SMD) and other clinical indicators. We constructed a predictive nomogram using R software (version 4.2.0; R Foundation for Statistical Computing, Vienna, Austria) and the “rms” package. Nomogram performance was evaluated through calibration plots. We assessed discriminative ability using Harrell’s concordance index (C-index) and time-dependent receiver operating characteristic (t-ROC) analyses. All statistical tests were two-sided, with *p* < 0.05 considered significant. Analyses were performed using R statistical software.

## 3. Results

### 3.1. Baseline Patient Characteristics

Of the 446 HCC patients included in the study, the mean age was 62.0 years, and 320 (71.7%) were male (Table 1). Key comorbidities included diabetes (28.3%) and hypertension (27.8%). Assessment of body composition found sarcopenia in 32.3% and myosteatosis in 41.5% of the patients. The etiology was primarily hepatitis B (47.8%) and hepatitis C (21.3%). Important laboratory values included a median total bilirubin of 0.86 mg/dL and a mean albumin of 3.5 g/dL. Tumor characteristics showed that 45.3% had a maximal tumor diameter >5 cm, and 26.5% had more than three tumors. The BCLC stages were mostly B (67.9%). The majority were classified as Child–Pugh class A (81.4%) and ALBI grade 2 (64.8%). The median number of TACE sessions was 2.

### 3.2. Overall Survival Analysis

The median follow-up period was 20.1 months (IQR, 17.5, 24.7). The 1-year, 3-year, and 5-year OS rates for the entire cohort were 67.7%, 33.0%, and 22.4%, respectively. Univariate analysis showed that albumin, the ALBI grade, AFP, up-to-seven criteria, and preoperative sarcopenia and myosteatosis were associated with OS (all *p* < 0.05) (Table 2). In multivariate analysis, albumin and sarcopenia were not independent prognostic factors for OS (all *p* > 0.05). However, the preoperative myosteatosis (hazard ratio [HR], 1.41; 95% confidence interval [CI], 1.14–1.75; *p* = 0.002), ALBI grade 2 (HR, 1.96; 95% CI, 1.53–2.53; *p* < 0.001), and ALBI grade 3 (HR, 3.37; 95% CI, 2.22–5.12; *p* < 0.001) remained independent prognostic factors.

### 3.3. Correlation Between Myosteatosis and Clinical Parameters

The median OS time for the cohort (*n* = 446) was 20.1 months (95% CI 17.5–24.7). The 1-, 2-, and 3-year OS rates were 67.7%, 45.5%, and 32.9%, respectively. Patients with myosteatosis had a lower OS time than patients without myosteatosis (16.1 vs. 25.6 months, *p* = 0.005). Patients with myosteatosis also had a lower PFS time than patients without myosteatosis (4.7 vs. 8.6 months, *p* = 0.010). The median PFS time for the cohort was 7.3 months (95% CI 6.1–8.9). The 1-, 2-, and 3-year PFS rates were 35.4%, 17.9%, and 12.8%, respectively.

The correlation analysis results indicate that SMD, measured in HU, was positively correlated with the SMI (r = 0.493, *p* < 0.001) and negatively correlated with the BMI (r = −0.290, *p* < 0.001) (Figure 1). Additionally, there was no significant relationship between SMD and albumin (*p* = 0.590) or between SMD and the ALBI grade (*p* = 0.684).

### 3.4. Establishment of the Prognostic Score Based on Myosteatosis and ALBI Grade

As illustrated in the Kaplan–Meier curves, both preoperative myosteatosis and ALBI grade 2–3 were significantly associated with worse overall survival (OS) (*p* = 0.005 and *p* < 0.001, respectively) (Figure 2A,B). To further discriminate between patients with different outcomes, we combined myosteatosis and the ALBI grade to generate six subgroups (Figure 2C). Patients with ALBI grade 3 and myosteatosis had the worst survival, whereas patients with ALBI grade 1, with or without myosteatosis, had the longest survival (*p* < 0.001). However, the survival of patients with ALBI grade 2 and myosteatosis was similar to that of patients with ALBI grade 3 without myosteatosis (*p* = 0.951), which led us to combine these two subgroups. Finally, we established the Myo-ALBI grade by combining myosteatosis and the ALBI grade, which was defined as follows: patients with ALBI grade 1 with or without myosteatosis were assigned a score of 0; patients with ALBI grade 2 without myosteatosis were assigned a score of 1; patients with ALBI grade 2 with myosteatosis or ALBI grade 3 without myosteatosis were assigned a score of 2; and patients with ALBI grade 3 with myosteatosis were assigned a score of 3 (Figure 2D).The relationships between clinicopathological factors and the Myo-ALBI grade are given in Table 3. There were 123 patients (24.7%) in the Myo-ALBI Grade 0 group, 172 (34.5%) in the Myo-ALBI Grade 1 group, 137 (27.5%) in the Myo-ALBI Grade 2 group, and 14 (2.8%) in the Myo-ALBI Grade 3 group. An increased Myo-ALBI grade was significantly associated with older age, female sex, higher BMI, hypertension, lower platelet count, higher INR, more advanced disease characteristics (higher tumor number and larger tumor size), and the presence of sarcopenia (all *p* < 0.05). These findings indicate that higher Myo-ALBI grades were linked with more severe disease and poorer liver function. Interestingly, patients with Myo-ALBI grade 1 often showed inverse correlations compared to grade 0 patients for several characteristics, including age, hypertension status, HBV carrier status, platelet count, and presence of sarcopenia.

### 3.5. Cumulative OS Rates According to the Myo-ALBI Grade

The median OS in the entire cohort (*n* = 446) was 20.1 months (95% CI, 17.5–24.7). The median OS values for the Myo-ALBI grade 0 (*n* = 123), Myo-ALBI grade 1 (*n* = 172), Myo-ALBI grade 2 (*n* = 137), and Myo-ALBI grade 3 (*n* = 14) groups were 33.9 months (95% CI, 23.7–44.2), 26.9 months (95% CI, 20.1–31.8), 13.8 months (95% CI, 11.6–17.5), and 7.2 months (95% CI, 5.4–11.8), respectively. The Kaplan–Meier curves for the 1-, 3-, and 5-year OS rates were divided into four groups according to the Myo-ALBI grade: Myo-ALBI grade 0: 82.1%, 47.1%, 34.3%; Myo-ALBI grade 1: 72.1%, 37.8%, 19.8%; Myo-ALBI grade 2: 55.5%, 17.4%, 11.3%; and Myo-ALBI grade 3: 7.1%, 0%, 0% (log-rank test: *p* < 0.001). Multivariate analyses reveal that the Myo-ALBI grade was an independent prognostic factor for OS (Table 4). Compared with Myo-ALBI grade 0, the adjusted risk for all-cause mortality was significantly higher in Myo-ALBI grade 1 (adjusted HR 1.63, 95% CI 1.24–2.15), Myo-ALBI grade 2 (adjusted HR 2.44, 95% CI 1.83–3.24), and Myo-ALBI grade 3 (adjusted HR 5.93, 95% CI 3.28–10.72). The risk of all-cause mortality in Myo-ALBI grade 3 was significantly higher than in the other groups (adjusted HR 3.52, 95% CI 2.02–6.15; *p* < 0.001). Other independent risk factors included a high AFP level (>200 ng/mL) and beyond the up-to-seven criteria (*p* = 0.001 and *p* < 0.001, respectively).

Furthermore, we explored the prognostic accuracies of the Myo-ALBI grade and each of its components—myosteatosis and the ALBI grade—using area under the curve (AUC) for the prediction of 3-year OS. The AUC results for Myo-ALBI grade, myosteatosis, and the ALBI grade were 0.665 (95% CI, 0.616–0.714), 0.559 (95% CI, 0.511–0.607), and 0.620 (95% CI, 0.573–0.666), respectively. According to the Z-test, the AUC for the Myo-ALBI grade was significantly higher than the presence of myosteatosis (*p* < 0.05) and superior, but not statistically significantly different, compared to the ALBI grade (*p* = 0.192).

### 3.6. Predictive Nomogram Based on the Myo-ALBI Grade

To provide a quantitative method for better outcome prediction, we constructed a nomogram that integrated the proven independent prognostic factors: the Myo-ALBI grade, AFP level, and up-to-seven criteria (Figure 3A). For internal validation, calibration plots of the nomogram predicting 2- and 3-year survival after 500 bootstrap resamples illustrated good agreement between predicted and observed survival outcomes (Figure 3B,C). The C-index of the nomogram based on the Myo-ALBI grade (Myo-ALBI nomogram: 0.743; 95% CI, 0.739–0.748) was significantly higher than the prognostic model without the Myo-ALBI grade (non-Myo-ALBI nomogram: 0.677; 95% CI, 0.673–0.682; *p* < 0.001), up-to-seven criteria (0.653; 95% CI, 0.649–0.657; *p* < 0.001), ALBI grade (0.616; 95% CI, 0.611–0.620; *p* = 0.008), and Child–Pugh class (0.573; 95% CI, 0.570–0.576; *p* < 0.001).

We further assessed the discriminative ability of the five prognostic models using time-dependent receiver operating characteristic (t-ROC) curves. Our analysis reveals that the nomogram incorporating the Myo-ALBI grade exhibited superior prognostic accuracy compared to the other four models (including the non-Myo-ALBI nomogram, up-to-seven criteria, ALBI grade, and Child–Pugh classification) throughout the study period for the entire patient population (Figure 4A). Notably, this improved predictive performance of the Myo-ALBI nomogram persisted when we focused our analysis on the BCLC-B subgroup of patients (Figure 4B). 

## 4. Discussion

Several studies have shown that both myosteatosis and the ALBI grade are associated with the prognosis of multiple cancers; however, most studies have addressed them individually. The correlations between myosteatosis and the ALBI grade and their combined prognostic values have remained unclear in HCC. In this study, we proved that myosteatosis and the ALBI grade were independent predictors of OS for patients with unresectable HCC undergoing TACE. Our combination of the Myo-ALBI grade and pretreatment clinicopathological data (the AFP level and up-to-seven criteria) demonstrated a modest performance in differentiating survival outcomes and exhibited superior prognostic ability for long-term survival compared to the Child–Pugh class and up-to-seven criteria. In summary, this study highlights the promising prognostic value of combining myosteatosis and the ALBI grade to assess HCC patients undergoing TACE in a clinical setting.

The exact mechanism through which the Myo-ALBI grade predicts the prognosis of HCC is not fully understood, but it likely stems from the integration of myosteatosis status with serum albumin and total bilirubin levels. Myosteatosis, characterized by the infiltration of fat into skeletal muscle, leading to reduced muscle quality and strength, is increasingly recognized as a poor prognostic factor in malignancies and is closely associated with tumor progression and mortality [20,21,22]. Specifically, myosteatosis has been shown to contribute to a poorer response to TACE and an increased risk of mortality in HCC, more so than sarcopenia [17]. This could be attributed to adipocytes in myosteatosis releasing inflammatory adipokines, which impair muscle blood flow and exacerbate insulin resistance. These changes may weaken the body’s immunity and promote cancer growth, leading to adverse treatment outcomes [23,24]. Furthermore, albumin and bilirubin levels play a pivotal role in predicting the prognosis of HCC, as they reflect liver synthetic function (albumin) and the liver’s ability to process waste (bilirubin). Changes in albumin and bilirubin levels can reveal the degree of liver damage and significantly impact outcomes for patients with HCC [25,26]. 

The Myo-ALBI grade, which encompasses both liver functional reserve, as indicated by albumin and bilirubin levels, and CT-derived muscle quality and nutritional status, as denoted by the presence or absence of preoperative myosteatosis, stands as a reliable tool for predicting outcomes. This integrated approach effectively identifies more at-risk HCC patients and provides a comprehensive assessment of prognostic factors. This is particularly crucial for patients with HCC undergoing TACE due to the treatment’s potential systemic effects on overall patient health. The prognostic accuracy of the Myo-ALBI grade was higher than either preoperative myosteatosis or the ALBI grade. Furthermore, patients with ALBI grade 3 and myosteatosis (Myo-ALBI grade 3) had a nearly fourfold increased risk of death from any cause compared to other groups. Patients with Myo-ALBI grade 3 had a median OS of 7.2 months (95% CI, 5.4–11.8), which is comparable to patients with BCLC-C HCC who received sorafenib treatment [27]. Therefore, caution is advised when considering TACE as a treatment option for these patients. Alternative treatment options with more favorable safety profiles and better survival benefits should be considered in this group.

The strong associations between Myo-ALBI grade 3 and various poor prognostic features (such as older age, higher BMI, elevated INR, lower albumin, higher bilirubin, and higher tumor burden) (Table 3) further validate this novel grading system. By combining myosteatosis and the ALBI grade, Myo-ALBI captures a broader spectrum of adverse factors than either component alone, potentially explaining its superior prognostic performance. The inverse relationship between Myo-ALBI grades 0 and 1 for several parameters highlights the complex interplay between liver function and muscle health in HCC. For instance, younger patients may progress to grade 1 earlier due to more aggressive disease, while older grade 0 patients may have developed compensatory mechanisms. The higher prevalence of HBV carriers in grade 0 could reflect earlier diagnosis through surveillance programs. Notably, the lower incidence of sarcopenia in grade 1 suggests muscle mass loss may precede changes in muscle quality (myosteatosis) in some patients. These observations underscore HCC’s complex pathophysiology and the value of Myo-ALBI as a comprehensive prognostic tool for clinical decision-making in HCC patients undergoing TACE.

In the context of BCLC stage B, TACE is the primary treatment for unresectable HCC patients [4]. However, prognoses can vary among HCC patients with the same BCLC-B stage due to differences in tumor size, number, vascularity, and underlying liver function [5,6]. According to the latest BCLC system, BCLC-B HCC can be divided into three subgroups. Patients with diffuse, infiltrative, or bilobar disease are unlikely to benefit from TACE, and systemic therapy should be considered instead [4]. However, there are no definite criteria for identifying BCLC-B HCC patients who should proceed directly to systemic treatment as their first-line therapy. In most major interdisciplinary treatment centers and clinical trials, HCC patients classified as Child–Pugh class A are often the primary focus [28]. This reflects the selection of a patient population with relatively preserved liver function, which is crucial for optimal treatment outcomes. The up-to-seven criteria, which combines the number of tumors and the size of the largest tumor, is an effective tool to stratify BCLC-B HCC patients for appropriate treatment modalities [11]. Patients within the up-to-seven criteria (limited tumor burden) benefit more from TACE, while those beyond the criteria should be considered for systemic therapies to improve survival outcomes [29,30]. Our Myo-ALBI nomogram was constructed by incorporating the Myo-ALBI grade, AFP level, and up-to-seven criteria, and it performed well in internal validation. When assessing overall survival in BCLC-B patients, the Myo-ALBI nomogram demonstrated higher predictive accuracy compared to non-Myo-ALBI nomograms, up-to-seven criteria, ALBI grade, and Child–Pugh class. Thus, the Myo-ALBI grade can improve the prediction of prognoses in patients with HCC undergoing TACE. In clinical practice, the Myo-ALBI grade can be used as a supplement to the Child–Pugh class or up-to-seven criteria to better stratify BCLC-B HCC patients and provide a more accurate basis for guiding postoperative follow-up and treatment.

While TACE primarily targets HCC, its efficacy and tolerability are linked to both liver function and overall patient health. The Myo-ALBI score, combining muscle quality (myosteatosis) and liver function (the ALBI grade), provides a comprehensive evaluation of a patient’s physiological reserve. This is particularly relevant for TACE due to its impact on treatment tolerability, recovery between sessions, ability to handle systemic effects, and potential influence on the tumor microenvironment [4,5,6]. As demonstrated in our previous work [17], myosteatosis is associated with poor TACE response and reduced survival. The underlying mechanism involves several factors, including inflammatory adipokines, impaired muscle blood flow, and insulin resistance. These factors can weaken immunity, promote cancer growth, and create a pro-tumorigenic environment, potentially influencing TACE effectiveness and tumor progression [17].

The Myo-ALBI score represents a significant advancement in prognostication for HCC patients considered for TACE. Unlike our previous SMAART score [30], which assessed patients after the first TACE session, the Myo-ALBI score is applied pre-TACE, allowing for more informed initial treatment decisions. Moreover, it uniquely incorporates myosteatosis, reflecting both hepatic and overall physical status. This comprehensive pre-treatment evaluation not only improves prognostic accuracy but also opens new avenues for personalized treatment strategies, potentially guiding decisions toward alternative or combination therapies in high-risk patients before TACE initiation.

Several limitations were present in our study. First, the retrospective design possibly introduced potential biases. Second, since the study was conducted at a single center, the generalizability of the findings may be limited, which underscores the need for larger-scale, multicenter randomized controlled trials to validate these results. Finally, the predominance of HBV infection as the cause of HCC in this Thai cohort may not reflect the etiology of HCC in other regions where HCV infection and alcohol abuse might be more prevalent. This regional variation could lead to differences in tumor imaging characteristics that potentially affect the applicability of our findings to broader populations.

## 5. Conclusions

Our study is the first to demonstrate that combining the presence of preoperative myosteatosis with the ALBI grade results in a novel and easily obtainable prognostic score, which is named the Myo-ALBI grade in this study. This score effectively predicts the prognosis of unresectable HCC patients undergoing TACE and enhances the prognostic value of the ALBI grade and up-to-seven criteria. However, external validation of the Myo-ALBI grade is needed to establish its efficacy and reliability in clinical practice.

## Figures and Tables

**Figure 1 cancers-16-03503-f001:**
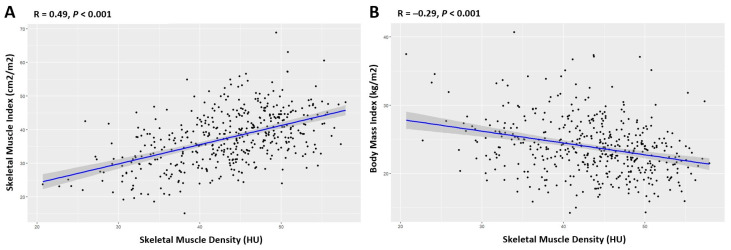
Association between the skeletal muscle density and clinical parameters. Scatter plots demonstrate that skeletal muscle density is positively associated with sarcopenia (**A**) while negatively correlated with body mass index (**B**).

**Figure 2 cancers-16-03503-f002:**
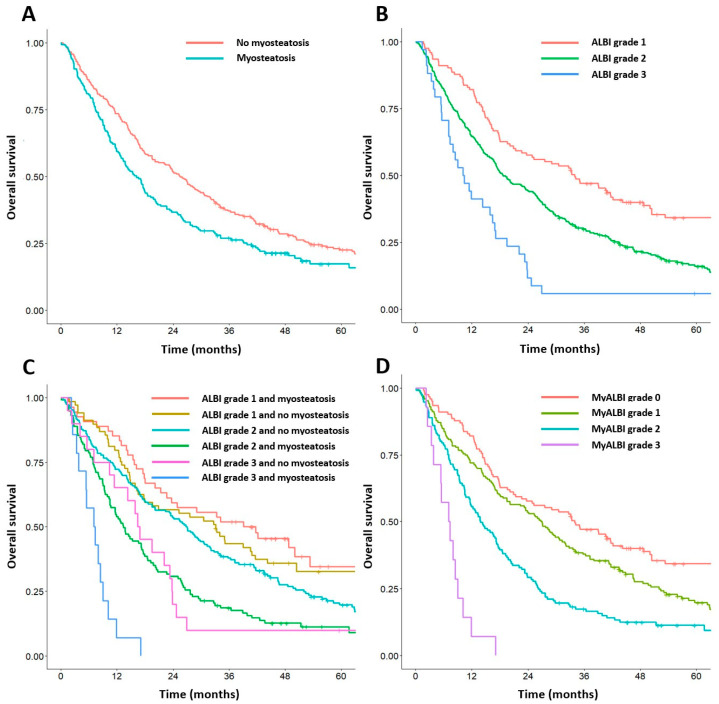
Kaplan–Meier analysis for overall survival of patients with hepatocellular carcinoma according to preoperative myosteatosis and ALBI grade. Kaplan–Meier analysis for OS according to (**A**) preoperative myosteatosis, (**B**) ALBI grade, (**C**) combination of preoperative ALBI grade and myosteatosis, and (**D**) Myo-ALBI grade. ALBI grade, albumin–bilirubin grade.

**Figure 3 cancers-16-03503-f003:**
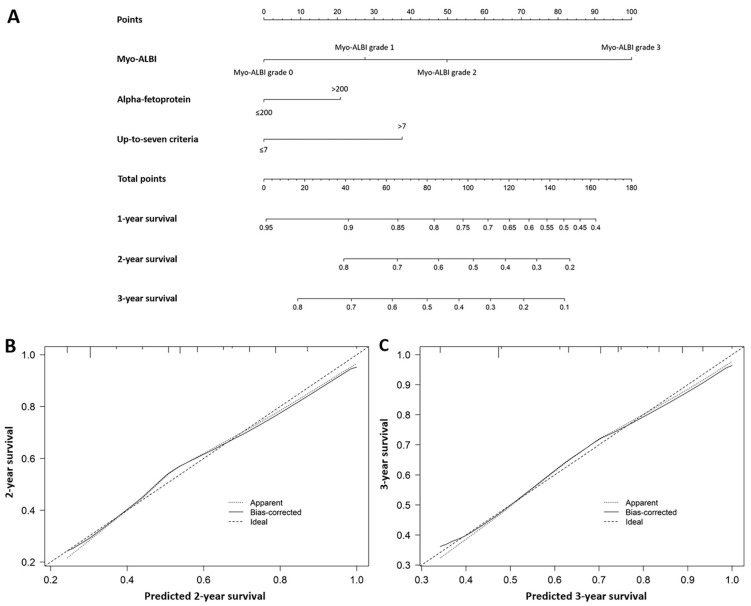
Nomogram for predicting 1-, 2-, and 3-year overall survival in hepatocellular carcinoma patients undergoing TACE. (**A**): Nomogram to predict 1-, 2-, and 3-year overall survival in HCC patients who underwent TACE. Calibration plot of the nomogram for 2-year survival (**B**) and 3-year survival (**C**). ALBI grade, albumin–bilirubin grade.

**Figure 4 cancers-16-03503-f004:**
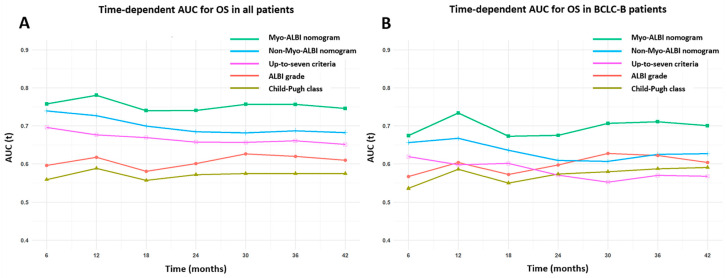
Time-dependent receiver operating characteristic curves for the Myo-ALBI nomogram, non-Myo-ALBI nomogram, up-to-seven criteria, ALBI grade, and Child–Pugh class for the prediction of overall survival in all patients (**A**) and BCLC-B patients (**B**). AUC, area under the curve; ALBI grade, albumin–bilirubin grade.

**Table 1 cancers-16-03503-t001:** Baseline characteristics of the entire hepatocellular carcinoma cohort.

Baseline Characteristics	All Cohorts (*n* = 446)
Age (years, mean ± SD)	62.0 ± 10.9
Male, *n* (%)	320 (71.7)
Comorbidity, *n* (%)	
Diabetes	126 (28.3)
Hypertension	124 (27.8)
Cardiovascular disease	29 (6.5)
Chronic kidney disease	17 (3.8)
Body composition	
BMI (kg/m^2^, <20.0/20.0–24.9/≥25.0), *n* (%)	75 (16.8)/220 (49.3)/151 (33.9)
SMI (cm^2^/m^2^, median (IQR))	37.5 (31.9, 43.44)
Sarcopenia, *n* (%)	144 (32.3)
SMD (HU, median (IQR))	44.3 (38.6, 48.7)
Myosteatosis, *n* (%)	185 (41.5)
Etiology, *n* (%)	
Hepatitis B	213 (47.8)
Hepatitis C	95 (21.3)
Alcohol	49 (11.0)
Others	89 (19.9)
Laboratory values	
Total bilirubin (mg/dL, median (IQR))	0.86 (0.53, 1.34)
Albumin (g/dL, mean ± SD)	3.5 ± 0.5
Platelet (×10^3^/L, median (IQR))	121 (77, 194)
International normalized ratio, median (IQR)	1.2 (1.1, 1.3)
Alpha-fetoprotein (ng/mL, <20/20–200/>200), *n* (%)	189 (42.4)/111 (24.9)/146 (32.7)
Tumor characteristics	
Maximal tumor diameter (<2/2–5/>5 cm), *n* (%)	57 (12.8)/187 (41.9)/202 (45.3)
Number of tumors (1/2/3/>3), *n* (%)	191 (42.8)/79 (17.7)/58 (13.0)/118 (26.5)
BCLC staging (A/B/C), *n* (%)	122 (27.4)/303 (67.9)/21 (4.7)
Models	
Child–Pugh class (A/B7/B8–9), *n* (%)	363 (81.4)/47 (10.5)/36 (8.1)
ALBI grade (1/2/3), *n* (%)	123 (27.6)/ 289 (64.8)/ 34 (7.6)
TACE sessions, median (IQR)	2 (1, 4)

ALBI, albumin–bilirubin; SD, standard deviation; BMI, body mass index; SMI, skeletal muscle index; SMD, skeletal muscle density; HU, Hounsfield units; IQR, interquartile range; BCLC, Barcelona Clinic Liver Cancer; TACE, transarterial chemoembolization.

**Table 2 cancers-16-03503-t002:** Univariate and multivariate analysis of clinicopathologic variables in relation to overall survival in hepatocellular carcinoma patients undergoing TACE (*n* = 446).

Clinicopathological Factors	Reference	Univariate Analysis		Multivariate Analysis	
		Hazard Ratio (95% CI)	*p*-Value	Hazard Ratio (95% CI)	*p*-Value
Age > 65 years	≤65 years	0.99 (0.80–1.22)	0.925		
Gender: female	male	0.97 (0.77–1.22)	0.819		
BMI: ≥25.0 kg/m^2^	<25.0 kg/m^2^	0.92 (0.74–1.14)	0.452		
Diabetes: yes	no	0.88 (0.70–1.10)	0.258		
Hypertension: yes	no	1.13 (0.90–1.42)	0.294		
Cardiovascular disease: yes	no	1.07 (0.72–1.60)	0.727		
Chronic kidney disease: yes	no	0.69 (0.39–1.22)	0.198		
Hepatitis B virus carrier: yes	no	1.05 (0.86–1.29)	0.635		
Hepatitis C virus carrier: yes	no	1.05 (0.82–1.33)	0.708		
Non-viral hepatitis: yes	no	0.89 (0.71–1.12)	0.311		
Platelet: <100 (×10^4^ µL)	≥100	0.94 (0.76–1.16)	0.550		
INR: >1.2	≤1.2	1.15 (0.93–1.41)	0.195		
Creatinine: >1.2 mg/dL	≤1.2 mg/dL	1.24 (0.89–1.74)	0.205		
Albumin: <4.0 g/dL	≥4.0 g/dL	1.69 (1.27–2.25)	<0.001		
Bilirubin: >2.0 mg/dL	≤2.0 mg/dL	1.37 (1.00–1.89)	0.054		
ALBI grade	Grade 1				
Grade 2		1.78 (1.39–2.29)	<0.001	1.96 (1.53–2.53)	<0.001
Grade 3		3.28 (2.17–4.96)	<0.001	3.37 (2.22–5.12)	<0.001
Alpha-fetoprotein: >200 ng/mL	≤200 ng/mL	1.68 (1.36–2.09)	<0.001	1.47 (1.18–1.83)	<0.001
Up-to-seven criteria: within	beyond	2.11 (1.71–2.59)	<0.001	2.00 (1.61–2.47)	<0.001
Sarcopenia: present	Absent	1.24 (1.00–1.54)	0.049		
Myosteatosis: present	Absent	1.34 (1.09–1.66)	0.006	1.41 (1.14–1.75)	0.002

TACE, transarterial chemoembolization; CI, confidence interval; BMI, body mass index; INR, International Normalized Ratio; ALBI, albumin–bilirubin.

**Table 3 cancers-16-03503-t003:** Relationships between the Myo-ALBI grade and clinicopathological characteristics in hepatocellular carcinoma patients undergoing TACE.

Clinicopathological Factors	Myo-ALBI Grade				
	Grade 0 (*n* = 123)	Grade 1 (*n* = 172)	Grade 2 (*n* = 137)	Grade 3 (*n* = 14)	*p*-Value
Age, year, mean (SD)	61.9 (11.7)	59.1 (9.8)	65.4 (10.6)	67.1 (10.7)	<0.001
Sex, *n* (%)					0.046
Male	96 (78.0)	128 (74.4)	87 (63.5)	9 (64.3)	
Female	27 (22.0)	44 (25.6)	50 (36.5)	5 (35.7)	
BMI, kg/m^2^, median (IQR)	23.1 (20.9, 26.1)	23.3 (20.8, 25.1)	24.5 (21.9, 27.3)	24.0 (22.5, 26.3)	0.032
Diabetes, *n* (%)					0.895
No	89 (72.4)	126 (73.3)	95 (69.3)	10 (71.4)	
Yes	34 (27.6)	46 (26.7)	42 (30.7)	4 (28.6)	
Hypertension, *n* (%)					<0.001
No	83 (67.5)	142 (82.6)	86 (62.8)	11 (78.6)	
Yes	40 (32.5)	30 (17.4)	51 (37.2)	3 (21.4)	
Cardiovascular disease, *n* (%)					0.064
No	109 (88.6)	163 (94.8)	131 (95.6)	14 (100)	
Yes	14 (11.4)	9 (5.2)	6 (4.4)	0 (0)	
Chronic kidney disease, *n* (%)					0.498
No	116 (94.3)	165 (95.9)	134 (97.8)	14 (100)	
Yes	7 (5.7)	7 (4.1)	3 (2.2)	0 (0)	
Hepatitis B virus carrier, *n* (%)					0.005
No	62 (50.4)	73 (42.4)	85 (62.0)	9 (64.3)	
Yes	61 (49.6)	99 (57.6)	52 (38.0)	5 (35.7)	
Hepatitis C virus carrier, *n* (%)					0.167
No	101 (82.1)	134 (77.9)	104 (75.9)	8 (57.1)	
Yes	22 (17.9)	38 (22.1)	33 (24.1)	6 (42.9)	
Platelet, ×10^3^/mm^3^, median (IQR)	186 (138.5, 233)	89.5 (68.8, 148.8)	114 (75, 171)	97 (65.8, 140.2)	<0.001
INR, median (IQR)	1.1 (1, 1.1)	1.2 (1.2, 1.4)	1.2 (1.1, 1.3)	1.3 (1.3, 1.4)	<0.001
Creatinine, median (IQR)	0.9 (0.8, 1.1)	0.9 (0.7, 1)	0.9 (0.7, 1.1)	0.8 (0.7, 1.1)	0.298
Albumin, ng/mL, median (IQR)	4.1 (3.9, 4.3)	3.4 (3.1, 3.6)	3.2 (2.9, 3.5)	2.6 (2.5, 2.7)	<0.001
Bilirubin, mg/dL, median (IQR)	0.5 (0.4, 0.6)	1 (0.7, 1.4)	1 (0.7, 1.6)	2.1 (1.4, 2.7)	<0.001
Alpha-fetoprotein, ng/mL, *n* (%)					0.261
>200	85 (69.1)	116 (67.4)	93 (67.9)	6 (42.9)	
≤200	38 (30.9)	56 (32.6)	44 (32.1)	8 (57.1)	
Size of largest tumor, cm, median (IQR)	5.4 (3.2, 9.2)	3.8 (2.6, 6.7)	4.7 (3.1, 9)	5.1 (3.9, 9)	0.011
Tumor number, *n*					<0.001
≤3	106 (86.2)	124 (72.1)	90 (65.7)	8 (57.1)	
>3	17 (13.8)	48 (27.9)	47 (34.3)	6 (42.9)	
Sarcopenia, *n* (%)					<0.001
Absent	80 (65.0)	140 (81.4)	76 (55.5)	6 (42.9)	
Present	43 (35.0)	32 (18.6)	61 (44.5)	8 (57.1)	

ALBI, albumin–bilirubin; TACE, transarterial chemoembolization; SD, standard deviation; BMI, body mass index; IQR, interquartile range; INR, International Normalized Ratio.

**Table 4 cancers-16-03503-t004:** Multivariate analysis of clinicopathologic variables in relation to overall survival in hepatocellular carcinoma patients undergoing TACE.

Clinicopathological Factors	Multivariate Analysis	
	Hazard Ratio (95% CI)	*p*-Value
Myo-ALBI grade		<0.001
Grade 0	Reference	
Grade 1	1.63 (1.24–2.15)	
Grade 2	2.44 (1.83–3.24)	
Grade 3	5.93 (3.28–10.72)	
Alpha-fetoprotein		0.001
≤200 ng/mL	Reference	
>200 ng/mL	1.45 (1.16–1.81)	
Up-to-seven criteria		<0.001
Within	Reference	
Beyond	1.96 (1.58–2.43)	

TACE, transarterial chemoembolization; CI, confidence interval; ALBI, albumin–bilirubin.

## Data Availability

All data were stored separately in a data repository and are available from the corresponding author on reasonable request.

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
