# Peer review of "Prognostic Value of Myosteatosis and Albumin–Bilirubin Grade for Survival in Hepatocellular Carcinoma Post Chemoembolization"

_cancers, 2024, doi:10.3390/cancers16203503_

Round 1

Reviewer 1 Report

Comments and Suggestions for Authors

My comments:

1.      This retrospective study demonstrated that both preoperative myosteatosis and ALBI grade 2-3 were significantly associated with worse overall survival (OS) (p = 0.005 and p <0.001, respectively). So they decided to established the Myo-ALBI grade by combining myosteatosis and the ALBI grade and demonstrated significant survival difference between grade 0,1,2 and 3.

2.      Though myosteatosis and ALBI grade all have been reported as prognostic markers for HCC patients, this study still could be considered to have novel findings.

3.      In Table 3, the. Relationships between the Myo-ALBI grade and various clinicopathological characteristics in HCC patients undergoing TACE were shown, which revealed the grade 3 patients significantly associated various poor prognostic features, such as older age, gender, BMI, INR, albumin, bilirubin, size of largest tumor and tumor number.

4.      However, for grade 1 patients, the correlations were often in a reverse way with grade 0 patients, such as age, hypertension, HBV carrier, platelet count and sarcopenia. Can the authors make some discussion for it?

5.      In addition, In Table 3, for the sarcopenia, it showed that grade 3 patients had the lowest proportion with sarcopenia. There must be some mistakes.

Comments on the Quality of English Language

The English is fluent.

Author Response

A detailed, point-by-point response to the reviewers' comments

Reviewer 1

Comment 1: This retrospective study demonstrated that both preoperative myosteatosis and ALBI grade 2-3 were significantly associated with worse overall survival (OS) (p = 0.005 and p <0.001, respectively). So they decided to established the Myo-ALBI grade by combining myosteatosis and the ALBI grade and demonstrated significant survival difference between grade 0,1,2 and 3.

Response 1: We appreciate the reviewer's accurate summary of our study's key findings. Indeed, our research demonstrated that both preoperative myosteatosis and ALBI grade 2-3 were independently associated with worse overall survival (OS) in patients with hepatocellular carcinoma undergoing transarterial chemoembolization (TACE).

The significant survival differences observed between patients with different combinations of these factors led us to develop the novel Myo-ALBI grade. This new prognostic tool combines the assessment of muscle quality (myosteatosis) with liver function (ALBI grade), providing a more comprehensive evaluation of patient status.

We believe this integrated approach offers several advantages:

  1. It captures both the systemic impact of cancer (reflected in muscle quality) and the degree of liver dysfunction.
  2. It allows for more nuanced patient stratification, potentially improving treatment decision-making.
  3. It demonstrates superior prognostic accuracy compared to either myosteatosis or ALBI grade alone.

Thank you for highlighting this crucial aspect of our study. We believe the Myo-ALBI grade represents a significant step forward in prognostic assessment for HCC patients undergoing TACE.

Comment 2: Though myosteatosis and ALBI grade all have been reported as prognostic markers for HCC patients, this study still could be considered to have novel findings.

Response 2: We appreciate the reviewer's recognition of the novelty in our study. While it's true that both myosteatosis and ALBI grade have been previously reported as individual prognostic markers for HCC patients, our study offers several novel contributions to the field:

  1. Combined prognostic model: To our knowledge, this is the first study to combine myosteatosis and ALBI grade into a single prognostic tool (Myo-ALBI grade) for HCC patients undergoing TACE. This integration provides a more comprehensive assessment of patient status, capturing both muscle quality and liver function in one metric.
  2. Synergistic effect: Our study demonstrates that the combination of these two factors provides superior prognostic value compared to either factor alone. This synergistic effect is a novel finding that enhances our understanding of prognostic factors in HCC.
  3. Specific to TACE: While previous studies have looked at these factors in various HCC populations, our study specifically focuses on patients undergoing TACE, providing valuable insights for this particular treatment modality.
  4. Development of a new nomogram: We have developed and internally validated a new nomogram incorporating the Myo-ALBI grade along with other clinical factors (AFP level and up-to-seven criteria), offering a practical tool for clinicians to estimate individual patient prognosis.
  5. Stratification of BCLC-B patients: Our findings provide a novel approach to further stratify BCLC-B HCC patients, potentially aiding in treatment decisions between TACE and systemic therapies.

We believe these novel aspects of our study contribute significantly to the field of HCC prognostication and may have important implications for clinical practice.

Comment 3: In Table 3, the. Relationships between the Myo-ALBI grade and various clinicopathological characteristics in HCC patients undergoing TACE were shown, which revealed the grade 3 patients significantly associated various poor prognostic features, such as older age, gender, BMI, INR, albumin, bilirubin, size of largest tumor and tumor number.

Response 3: We appreciate the reviewer's keen observation regarding the relationships between the Myo-ALBI grade and various clinicopathological characteristics as presented in Table 3. Indeed, our analysis reveals that patients with Myo-ALBI grade 3 are significantly associated with several poor prognostic features. This finding underscores the potential of the Myo-ALBI grade as a comprehensive prognostic tool. To further emphasize this point, we have made the following additions to our manuscript:

In the Discussion section (page 12, 2nd paragraph), we have included a new paragraph to contextualize these findings:

“The strong associations between Myo-ALBI grade 3 and various poor prognostic features (such as older age, higher BMI, elevated INR, lower albumin, higher bilirubin, and higher tumor burden) (Table 3) further validate this novel grading system. By combining myosteatosis and ALBI grade, Myo-ALBI captures a broader spectrum of adverse factors than either component alone, potentially explaining its superior prognostic performance.”

This finding underscores the potential of the Myo-ALBI grade as a comprehensive prognostic tool.

Comment 4: However, for grade 1 patients, the correlations were often in a reverse way with grade 0 patients, such as age, hypertension, HBV carrier, platelet count and sarcopenia. Can the authors make some discussion for it?

Response 4: We appreciate the reviewer's careful examination of the data in Table 3 and the insightful observation regarding the relationships between Myo-ALBI grades 0 and 1. We agree that this pattern deserves further discussion. To address this, we have added the following content to our manuscript:

In the Results section (page 7, 1st paragraph), we have added:

“Interestingly, patients with Myo-ALBI grade 1 often showed inverse correlations compared to grade 0 patients for several characteristics, including age, hypertension status, HBV carrier status, platelet count, and presence of sarcopenia.”

In the Discussion section (page 12, 2nd paragraph), we have included a new paragraph to address this observation:

“The inverse relationship between Myo-ALBI grades 0 and 1 for several parameters highlights the complex interplay between liver function and muscle health in HCC. For instance, younger patients may progress to grade 1 earlier due to more aggressive disease, while older grade 0 patients may have developed compensatory mechanisms. The higher prevalence of HBV carriers in grade 0 could reflect earlier diagnosis through surveillance programs. Notably, the lower incidence of sarcopenia in grade 1 suggests muscle mass loss may precede changes in muscle quality (myosteatosis) in some patients. These observations underscore HCC's complex pathophysiology and the value of Myo-ALBI as a comprehensive prognostic tool for clinical decision-making in HCC patients undergoing TACE.”

These additions aim to address the reviewer's comment by providing potential explanations for the observed patterns and acknowledging the complexity of the relationships between the Myo-ALBI grade and various clinical parameters. We thank the reviewer for this valuable observation, which has allowed us to deepen our discussion and highlight areas for future research.

Comment 5: In addition, In Table 3, for the sarcopenia, it showed that grade 3 patients had the lowest proportion with sarcopenia. There must be some mistakes.

Response 5: We sincerely appreciate the reviewer's careful examination of our data, particularly regarding the relationship between Myo-ALBI grade and sarcopenia as presented in Table 3. We acknowledge that there was an error in the original table, which we have now corrected. We thank the reviewer for bringing this to our attention.

To clarify:

  1. Data correction: We have thoroughly reviewed our original dataset and corrected the labeling for sarcopenia in Table 3. The "Absent" and "Present" rows were inadvertently switched in the original table.
  2. Corrected interpretation: With the accurate data, we can now confirm that grade 3 patients indeed had the highest proportion of sarcopenia (57.1%), followed by grade 2 (44.5%), grade 0 (35.0%), and grade 1 (18.6%).
  3. Consistency with expectations: This corrected data aligns better with clinical expectations, showing an increasing prevalence of sarcopenia with worsening Myo-ALBI grade, which is consistent with the progressive nature of liver disease and cancer-related muscle wasting.
  4. Implications: This correction strengthens the validity of our Myo-ALBI grading system, demonstrating its ability to capture the relationship between liver function, muscle health, and overall patient status.

We have updated Table 3 in our manuscript to reflect these corrections. We thank the reviewer for their astute observation, which has allowed us to improve the accuracy of our data presentation and strengthen the overall quality of our study. We are grateful for the opportunity to address this issue and provide more accurate information to our readers.

Reviewer 2 Report

Comments and Suggestions for Authors

The authors investigate how myosteatosis and liver function influence survival outcome in patients undergoing TACE, resulting in Myo-ALBI prognostic score but I would also correlate myosteatosis AND tumor progression or recurrence rate or recurrence free survival, as TACE is performed for HCC and not for cirrhosis. Otherwise, there is a linear concept correlation between Myo-ALBI & patient survival but what is the rationale to put TACE in this line, if TACE regards the tumor and not the liver function? This needs to be explained. 

Author Response

A detailed, point-by-point response to the reviewers' comments

Reviewer 2

Comment: The authors investigate how myosteatosis and liver function influence survival outcome in patients undergoing TACE, resulting in Myo-ALBI prognostic score but I would also correlate myosteatosis AND tumor progression or recurrence rate or recurrence free survival, as TACE is performed for HCC and not for cirrhosis. Otherwise, there is a linear concept correlation between Myo-ALBI & patient survival but what is the rationale to put TACE in this line, if TACE regards the tumor and not the liver function? This needs to be explained. 

Response: We appreciate the reviewer's insightful comment and the opportunity to clarify our approach. We agree that exploring the relationship between myosteatosis and tumor progression or recurrence provides valuable insights. To address this important point:

  1. In our previous work (reference 17), we have already demonstrated a significant correlation between myosteatosis and TACE response. Patients with myosteatosis had a lower rate of TACE response than those without (56.1% vs. 68.7%, adjusted odds ratio 0.49, 95% CI 0.34–0.72). This finding underscores the relevance of myosteatosis to tumor-specific outcomes in the context of TACE.
  2. To further strengthen our current study, we have added the following analyses to the Material and methods section (page 3, section 2.3):

“Progression-free survival (PFS) was calculated from the date of initial TACE session to the date of radiological progression according to mRECIST criteria, death from any cause, or the censoring date of June 30, 2023, whichever occurred first.”

Results section (page 6, section 3.3):

“The median OS time for the cohort (n = 446) was 20.1 months (95% CI 17.5–24.7). The 1-, 2-, and 3-year OS rates were 67.7%, 45.5%, and 32.9%, respectively. Patients with myosteatosis had a lower OS time than patients without myosteatosis (16.1 vs. 25.6 months, P = 0.005). Patients with myosteatosis also had a lower PFS time than patients without myosteatosis (4.7 vs. 8.6 months, P = 0.010). The median PFS time for the cohort was 7.3 months (95% CI 6.1–8.9). The 1-, 2-, and 3-year PFS rates were 35.4%, 17.9%, and 12.8%, respectively.”

  1. In the Discussion section (page 12, 4th paragraph), we have added a new paragraph to address the rationale for incorporating TACE in our analysis:

“While TACE primarily targets HCC, its efficacy and tolerability are linked to both liver function and overall patient health. The Myo-ALBI score, combining muscle quality (myosteatosis) and liver function (ALBI grade), provides a comprehensive evaluation of a patient's physiological reserve. This is particularly relevant for TACE due to its impact on treatment tolerability, recovery between sessions, ability to handle systemic effects, and potential influence on the tumor microenvironment [4-6]. As demonstrated in our previous work [17], myosteatosis is associated with poor TACE response and reduced survival. The underlying mechanism involves several factors, including inflammatory adipokines, impaired muscle blood flow, and insulin resistance. These factors can weaken immunity, promote cancer growth, and create a pro-tumorigenic environment, potentially influencing TACE effectiveness and tumor progression [17].”

These additions aim to address the reviewer's important points by providing data on tumor-specific outcomes, clarifying the rationale for considering both muscle health and liver function in the context of TACE for HCC, and maintaining the explanation of the underlying mechanisms. We thank the reviewer for this valuable feedback, which has allowed us to strengthen the clinical relevance and comprehensiveness of our study.
